# GRAPH TOPOLOGICAL FEATURES VIA GAN

## ABSTRACT

Inspired by the success of generative adversarial networks (GANs) in image domains, we introduce a novel hierarchical architecture for learning characteristic topological features from a single arbitrary input graph via GANs. The hierarchical architecture consisting of multiple GANs preserves both local and global topological features, and automatically partitions the input graph into representative stages for feature learning. The stages facilitate reconstruction and can be used as indicators of the importance of the associated topological structures. Experiments show that our method produces subgraphs retaining a wide range of topological features, even in early reconstruction stages. This paper contains original research on combining the use of GANs and graph topological analysis.

## 1 INTRODUCTION

Graphs have great versatility, able to represent complex systems with diverse relationships between objects and data. With the rise of social networking, and the importance of relational properties to the "big data" phenomenon, it has become increasingly important to develop ways to automatically identify key structures present in graph data. Identification of such structures is crucial in understanding how a social network forms, or in making predictions about future network behavior. To this end, a large number of graph analysis methods have been proposed to analyze network topology at the node (Muppidi & Koraganji, 2016), community (Fortunato, 2010; Martínez et al., 2016), and global levels (Wu et al., 2017).

Each level of analysis is greatly influenced by network topology, and thus far algorithms cannot be adapted to work effectively for arbitrary network structures. Modularity-based community detection (Xiang et al., 2016) works well for networks with separate clusters, whereas edge-based methods (Delis et al., 2016) are suited to dense networks. Similarly, when performing graph sampling, Random Walk (RW) is suitable for sampling paths (Leskovec & Faloutsos, 2006), whereas Forrest Fire (FF) is useful for sampling clusters (Leskovec et al., 2005). When it comes to graph generation, Watts-Strogatz (WS) graph models (Watts & Strogatz, 1998) can generate graphs with small world features, whereas Barabsi-Albert (BA) graph models (Barabási & Albert, 1999) simulate super hubs and regular nodes according to the scale-free features of the network.

However, real-world networks typically have multiple topological features. Considering real-world networks also introduces another issue that traditional graph analysis methods struggle with; having a mere single instance of a graph (e.g. the transaction graph for a particular bank), making it difficult to identify the key topological properties in the first place. In particular, we are interested in both "local topological features" (such as the presence of subgraph structures like triangles) and "global topological features" such as degree distribution.

Instead of directly analyzing the entire topology of a graph, GTI first divides the graph into several hierarchical layers. A hierarchical view of a graph can split the graph by local and global topological features, leading to a better understanding of the graph (Watts et al., 2002). As different layers have different topological features, GTI uses separate GANs to learn each layer and the associated features. By leveraging GANs renowned feature identification (Goodfellow et al., 2014) on each layer, GTI has the ability to automatically capture arbitrary topological features from a single input graph. Figure 1 demonstrates how GTI can learn to reproduce an input graph where a single GAN cannot.

In addition to learning topological features from the input graph, the GTI method defines a reconstruction process for reproducing the original graph via a series of reconstruction stages (the number

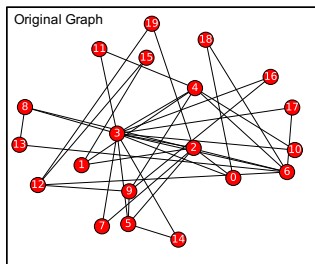 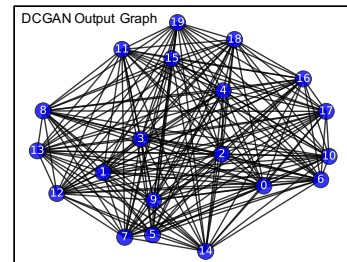 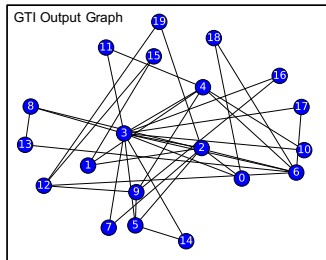

Figure 1: How GTI recovers the original graph while naive GAN methods do not: The DCGAN output looks like a complete graph, whereas GTI can capture the super-hub structure of node 3 and node 2.

of which is automatically learned during training). As stages are ranked in order of their contribution to the full topology of the original graph, early stages can be used as an indicator of the most important topological features. Our focus in this initial work is on the method itself and demonstrating our ability to learn these important features quickly (via demonstrating the retention of identifiable structures and comparisons to graph sampling methods).

## 2 METHOD

In this section, we demonstrate the work flow of GTI (see Figure 2), with a particular focus on the GAN, Sum-up and Stage Identification modules. At a high level, the GTI method takes an input graph, learns its hierarchical layers, trains a separate GAN on each layer, and autonomously combines their output to reconstruct stages of the graph. Here we give a brief overview of each module.

**Hierarchical Identification Module:** This module detects the hierarchical structure of the original graph using the Louvain hierarchical community detection method (Blondel et al., 2008), denoting the number of layers as $L$. The number of communities in each layer is used as a criterion for how many subgraphs a layer should pass to the next module.

**Layer Partition Module:** Here we partition a given layer into $M$ non-overlapping subgraphs, where $M$ is the number of communities. We do not use the learned communities from the Louvain method as we cannot constrain the size of any community. We instead balance the communities into fixed size subgraphs using the METIS approach (Karypis & Kumar, 1995).

**Layer GAN Module:** Rather than directly using one GAN to learn the whole graph, we use different GANs to learn features for each layer separately. If we use a single GAN to learn features for the whole graph, some topological features may be diluted or even ignored. This module regenerates subgraphs, with the Layer Regenerate Module generating the adjacency matrix for the corresponding layer. For more detail see Section 2.1.

**Layer Regenerate Module:** Here, for a given layer, the corresponding GAN has learned all the properties of each subgraph, meaning we can use the generator in this GAN to regenerate the topology of the layer by generating $M$ subgraphs of $k$ nodes. Note that this reconstruction only restores edges within each non-overlapping subgraph, and does not include edges between subgraphs.

**All Layer Sum-up Module:** This module outputs a weighted reconstructed graph by summing up all reconstructed layers along with the edges between subgraphs that were not considered in the Regenerate Module. The "weight" of each edge in this module represents its importance to the reconstruction. Indeed, we rely upon these weights to identify the reconstruction stages for the original graph. For more details, see Section 2.2.

**Stage Identification Module:** By analyzing the weighted adjacency matrix of the Sum-up Module, we extract stages for the graph. These stages can be interpreted as steps for a graph reconstruction process. See Section 2.3 for details.

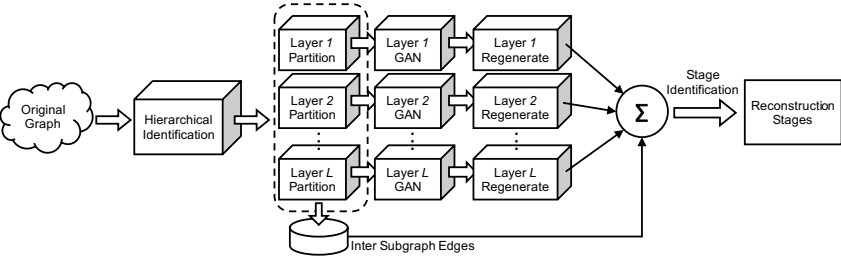

Figure 2: Work flow for GTI.

## 2.1 LAYER GAN MODULE

Figure 3(a) and Figure 3(b) show the architectures for the generator and discriminator of the GAN. Where the generator is a deconvolutional neural network with the purpose of restoring a $k \times k$ adjacency matrix from the standard uniform distribution, the discriminator is instead a CNN whose purpose is to estimate if the input adjacency matrix is from a real dataset or from a generator. Here, $BN$ is short for batch normalization which is used instead of max pooling because max pooling selects the maximum value in the feature map and ignores other values, whereas $BN$ will synthesize all available information. $LR$ stands for the leaky ReLU active function ($LR = max(x, 0.2 \times x)$) which we use since $0$ has a specific meaning for adjacency matrices. In addition, $k$ represents the size of a subgraph, and $FC$ the length of a fully connected layer. We set the stride for the convolutional/deconvolutional layers to be 2. We adopt the same loss function and optimization strategy (1000 iterations of ADAM (Kingma & Ba, 2014) with a learning rate of 0.0002) used in the DCGAN method of Radford et al. (2015).

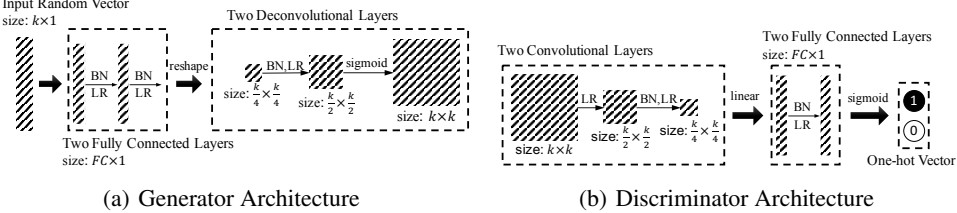

(a) Generator Architecture      (b) Discriminator Architecture

Figure 3: Generator and discriminator architecture

## 2.2 SUM-UP MODULE

In this module, we use a linear function (see Equation 1) to add the graphs from all layers together. $re_G$ is the reconstructed adjacency matrix (with input from all layers), $G'_i, i \in L$ is the reconstructed adjacency matrix for each layer (with $G$ representing the full original graph with $N$ nodes), $E$ refers to all the inter-subgraph (community) edges identified by the Louvain method from each hierarchy, and $b$ represents a bias. While each layer of the reconstruction may lose certain edge information, summing up the hierarchical layers along with $E$ will have the ability to reconstruct the entire graph.

$$re_G = \sum_{i=1}^{L} w_i G'_i + wE + b \qquad (1)$$

To obtain $w$ and $b$ for each layer, we use Equation 2 as the loss function (where we add $\epsilon = 10^{-6}$ to avoid taking $log(0)$ or division by 0), minimizing over 500 iterations of SGD with learning rate 0.1. We note that Equation 2 is similar to a $KL$ divergence, though of course $re_G$ and $G$ are not probability distributions.

$$\text{Loss}\,(re_G, G) = \sum_{i \in 1 \cdots N^2} vec(G + \epsilon)_i \cdot log \frac{vec(G + \epsilon)_i}{vec(re_G + \epsilon)_i} \qquad (2)$$

## 2.3 STAGE IDENTIFICATION

$re_G$ can be interpreted as representing how each edge contributes to the entire topology. According to these weights we can then divide the network into several stages, with each stage representing a collection of edges greater than a certain weight. We introduce the concept of a "cut-value" to turn $re_G$ into a binary adjacency matrix.

We observe that many edges in $re_G$ share the same weight, which implies these edges share the same importance. Furthermore, the number of unique weights can define different reconstruction stages, with the most important set of edges sharing the highest weight. Each stage will include edges with weights greater than or equal to the corresponding weight of that stage. Hence, we define an ordering of stages by decreasing weight, giving insight on how to reconstruct the original graph in terms of edge importance. We denote the $i$th largest unique weight-value as $CV_i$ (for "cut value") and thereby define the stages as in Equation 3 (an element-wise product), where $I[w \geq CV_i]$ is an indicator function for each weight being equal or larger than the $CV_i$.

$$re_G^i = re_G I[w \geq CV_i] \tag{3}$$

In Section 4, we use synthetic and real networks to show that each stage preserves identifiable topological features of the original graph during the graph reconstruction process. As each stage contains a subset of the original graphs edges, we can interpret each stage as a sub-sampling of the original graph. This allows us to compare with prominent graph sampling methodologies to emphasize our ability to retain important topological features.

## 3 RELATED WORK

The importance of deep learning and the growing maturity of graph topology analysis has led to more focus on the ability to use the former for the latter (Li et al., 2015). A number of supervised and semi-supervised learning methods have been developed for graph analysis. A particular focus is on the use of CNNs (Bruna et al., 2014; Henaff et al., 2015; Duvenaud et al., 2015; Defferrard et al., 2016). These new methods have shown promising results for their respective tasks in comparison to traditional graph analysis methods (such as kernel-based methods, graph-based regularization techniques, etc).

Since GANs were first introduced (Goodfellow et al., 2014), its theory and application has expanded greatly. Many advances in training methods (Denton et al., 2015; Chen et al., 2016; Zhao et al., 2016; Nowozin et al., 2016) have been proposed in recent years, and this has facilitated their use in a number of applications. For example, GANs have been used for artwork synthesis (Tan et al., 2017), text classification (Miyato et al., 2016), image-to-image translation (Yi et al., 2017), imitation of driver behavior (Kuefler et al., 2017), identification of cancers (Kohl et al., 2017), and more. The GTI method expands the use of GANs into the graph topology analysis area.

A distinguishing feature of our method is that it is an unsupervised learning tool (facilitated by the use of GANs) that leverages the hierarchical structure of a graph. GTI can automatically capture both local and global topological features of a network. To the best of the authors' knowledge, this is the first such unsupervised method.

## 4 EVALUATION

All experiments in this paper were conducted locally on CPU using a Mac Book Pro with an Intel Core i7 2.5GHz processor and 16GB of 1600MHz RAM. Though this limits the size of our experiments in this preliminary work, the extensive GAN literature (see Section 3) and the ability to parallelize GAN training based on hierarchical layers suggests that our method can be efficiently scaled to much larger systems.

### 4.1 DATASETS

We use a combination of synthetic and real datasets. Through the use of synthetic datasets with particular topological properties, we are able to demonstrate the retention of these easily identifiable

Table 1: Size of original datasets, and corresponding reconstruction stages

| Graph | # Nodes | # Edges | # Stages | Retained edge percentage for ordered stages (%) |
|---|---|---|---|---|
| BA | 500 | 996 | 7 | 19.48, 26.31, 36.04, 39.36, 41.57, 57.43, 100 |
| ER | 500 | 25103 | 4 | 4.32, 21.73, 94.91, 100 |
| Kronecker | 2178 | 25103 | 10 | 87.77, 88.65, 91.76, 91.89, 92.47, 93.32, 96.06, 97.05, 98.57, 100 |
| WS | 500 | 500 | 7 | 11.20, 11.40, 16.00, 18.00, 54.60, 97.80, 100 |
| Facebook | 4039 | 88234 | 7 | 52.28, 83.33, 87.49, 91.41, 90.31, 91.95, 100 |
| Wiki-Vote | 7115 | 103689 | 4 | 58.31, 73.79, 85.60, 100 |
| RoadNet | 5371 | 7590 | 12 | 0.62, 3.87, 26.64, 27.98, 31.79, 32.42, 34.22, 34.65, 34.80, 64.06, 76.81, 100 |
| P2P | 3334 | 6627 | 7 | 49.04, 53.90, 70.32, 87.54, 88.40, 89.65, 100 |

properties across the reconstruction stages. Of course, in real-world applications we do not know the important topological structures a priori, and so also demonstrate our method on a number of real-world datasets of varying sizes.

We use the ER graph model (Erdos & Rényi, 1960), the BA graph model (Barabási & Albert, 1999), the WS graph model (Watts et al., 2002) , and the Kronecker graph model (Leskovec et al., 2010) to generate our synthetic graphs. The varying sizes of our synthetic graphs (as well as our real-world datasets) are outlined in Table 1. The ER ($p = 0.2$), WS ($k = 2, p = 0.1$) and BA ($m = 2$) graphs were generated using the NetworkX package (Hagberg et al., 2008). The Kronecker graph was generated using the krongen package of the SNAP project[1].

For real datasets, we use data available from the Stanford Network Analysis Project (Leskovec, 2014). In particular, we use the Facebook network, the wiki-Vote network, and the P2P-Gnutella network. The Facebook dataset consists of "friends lists", collected from survey participants according to the connections between user-accounts on the online social network. It includes node features, circles, and ego networks; all of which has been anonymized by replacing the Facebook-internal ids. Wiki-vote is a voting network (who votes for whom etc) that is used by Wikipedia to elect page administrators; P2P-Gnutella is a peer-to-peer file-sharing network: Nodes represent hosts in the Gnutella network topology, with edges representing connections between the hosts. RoadNet is a connected component of the road network of Pennsylvania. Intersections and endpoints are represented by nodes, and the roads connecting them are edges.

## 4.2 Local Topological Features

Here we use two examples to demonstrate how GTI retains important local topological structure during each reconstruction stage.

**BA Network Stage Analysis:** We demonstrate the reconstruction process of a BA network in Figure 4, with the top row demonstrating the entire reconstruction process of the full network. We clearly observe that each reconstructed network becomes denser and denser as additional stages are added. The bottom row of Figure 4 shows the subgraphs corresponding to nodes 0 to 19 at each reconstruction stage. We observe that these subgraphs retain the most important feature of the original subgraph (the star structures at node 0), even during the first reconstruction stage.

**Road Network Stages Analysis:** We observe in Table 1 that the retained edge percentages of the RoadNet reconstruction decrease more consistently with each stage than in the BA network. This is reasonable, because geographical distance constraints naturally result in fewer super hubs, with each node having less variability in its degree. In Figure 5, we observe the reconstruction of the full network, and the node 0 to node 19 subgraph of RoadNet. We observe in the bottom row of Figure 5 that the dominant cycle structure of the original node 0-19 subgraph clearly emerges. We also observe an interesting property of the stages of the original graph in the top row of Figure 5. As SNAP does not provide the latitude and longitude of nodes, we cannot use physical location. We instead calculate the modularity of each stage, where modularity represents the tightness of the community (Newman, 2006). We found that the modularity deceases from 0.98 to 0.92 approximately linearly. This indicates that GTI views the dense connections between local neighborhoods as a particularly representative topological property of road networks, as such clusters are formed before links between clusters.

---

[1]https://github.com/snap-stanford/snap/tree/master/examples/krongen

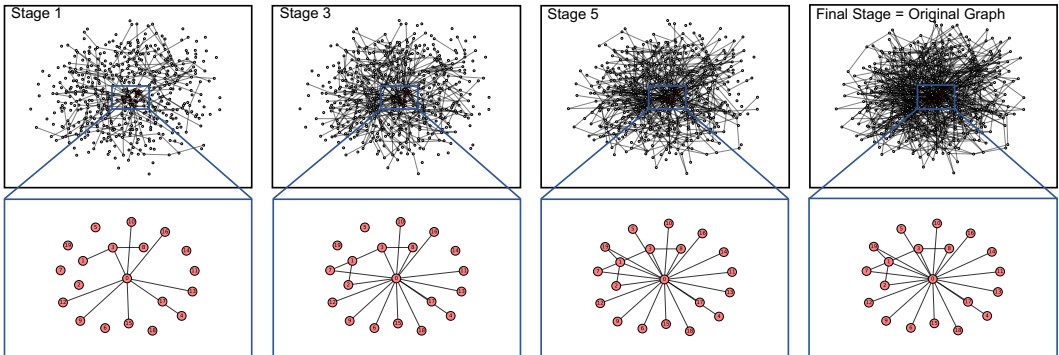

Figure 4: The topology of original graph and corresponding stages of BA network.

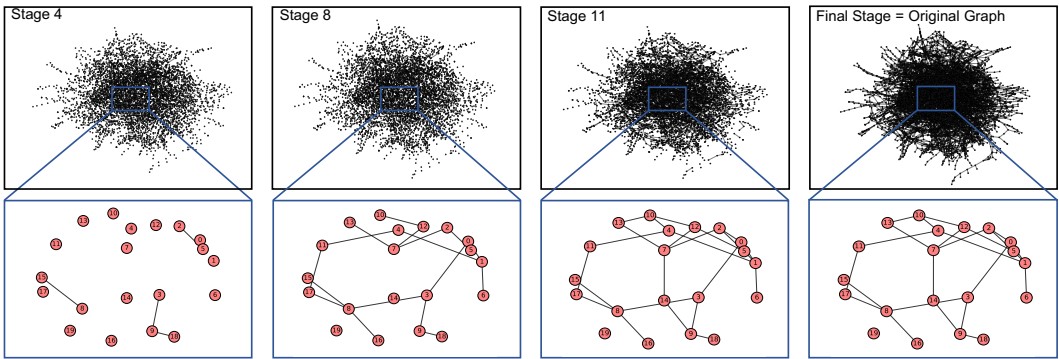

Figure 5: The topology of original graph and corresponding stages of road network.

### 4.3 GLOBAL TOPOLOGICAL FEATURES

In the previous section, we demonstrated GTI's ability to the preserve local topological features. Here we focus on degree distribution and the distribution of cluster coefficients, global topological features. Figure 6 and Figure 7 respectively show the log-log degree distributions and log-log cluster coefficient distributions for each of the datasets given in Table 1, where the horizontal axis represents the ordered degrees (ordered cluster coefficient), and the vertical axis the corresponding density. The red line is used to demonstrate the degree distribution (cluster coefficient distribution) of the original graph, with the distributions of the ordered stages represented by a color gradient from green to blue.

We observe that with the exception of the ER network, the degree distributions and the cluster coefficient distributions of early stages are similar to the original graphs, and only become more accurate as we progress through the reconstruction stages. Although the degree distributions and cluster coefficient distributions for the early stages of the ER network reconstruction are shifted, we observe that GTI quickly learns the Poisson like shape in degree distribution, and also learns the "peak-like" shape in the cluster coefficient distribution. This is particularly noteworthy given that the ER model has no true underlying structure (as graphs are chosen uniformly at random). Finally, we note that GTI quickly learns that the cluster coefficient of the WS network is zero.

### 4.4 COMPARISON WITH GRAPH SAMPLING

The graphs generated by GTI can be considered as samples of the original graph in the sense that they are representative subgraphs of a large input graph. We compare the performance of GTI with that of other widely used graph sampling algorithms (Random Walk, Forest Fire and Random Jump) with respect to the ability to retain topological structures (Leskovec & Faloutsos, 2006).

We demonstrate this through the subgraph structures of the BA and Facebook datasets, comparing stage 1 of GTI against the graph sampling algorithms (designed to terminate with the same number

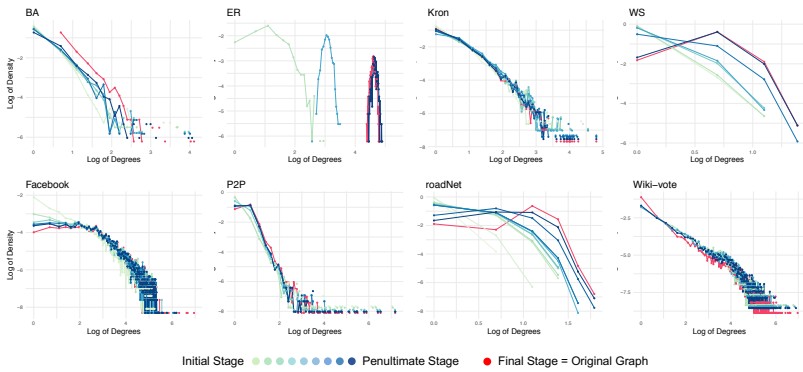

Figure 6: Degree distributions for 8 datasets.

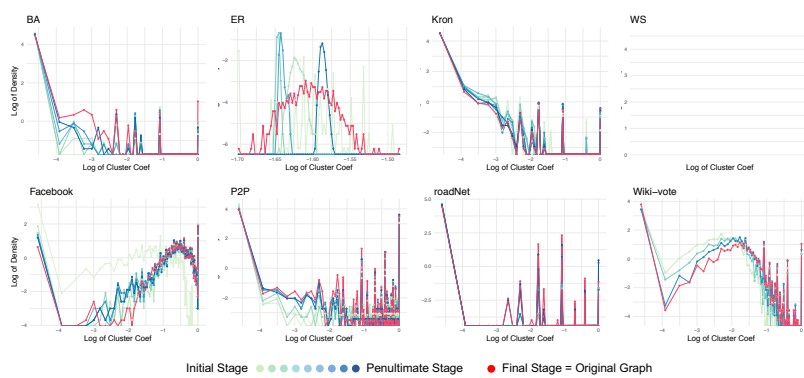

Figure 7: Cluster coefficient distributions for 8 datasets.

of nodes as the GTI stage). We take out each of subgraphs from BA and Facebook network (nodes 0-19 and nodes 0-49) to visually compare the ability of the first stage of GTI to retain topological features in comparison to the three graph sampling methods. In Figure 8, we observe that stage 1 of GTI has retained a similar amount of structure in the 20 node BA subgraph as Forest Fire (Leskovec et al., 2005), while demonstrating considerably better retention than either Random Walk or Random Jump. However, for 50 node BA subgraph, only GTI has the ability to retain the two super hubs present in the original graph. In Figure 9, we observe that GTI demonstrates vastly superior performance to the other methods when run on the Facebook dataset, which has a number of highly dense clusters with very sparse inter-cluster connections.

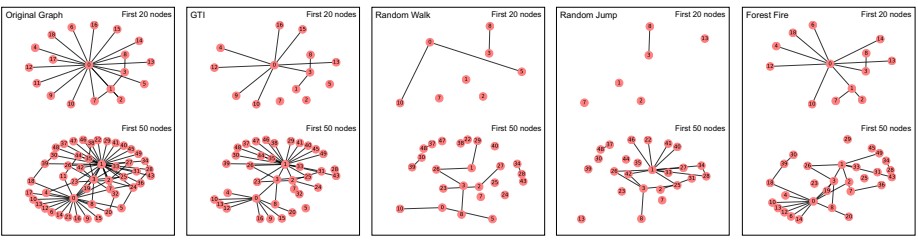

Figure 8: Comparison with graph sampling methods on the BA subgraphs.

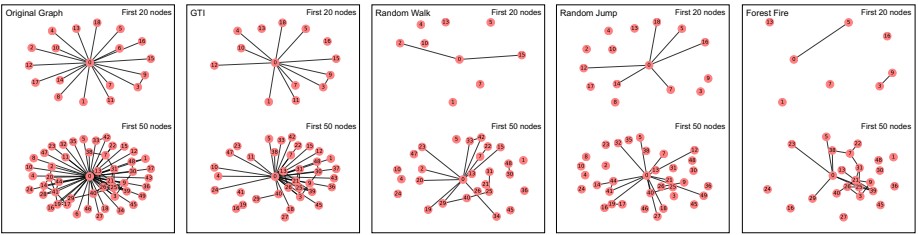

Figure 9: Comparison with graph sampling methods on the Facebook subgraphs.

Table 2: F-norm numerical evaluation on penultimate stage, corresponding graph sampling methods and base model

| Graph | GTI (penultimate stage) | Random Walk | Random Jump | Forest Fire | BaseModel |
|---|---|---|---|---|---|
| BA | 53.02 | 57.50 | 56.64 | 57.79 | 62.90±0.09 |
| ER | 285.86 | 232.68 | 231.05 | 232.64 | 282.93±0.56 |
| kronecker | 124.93 | 123.92 | 120.64 | 124.58 | 125.32±0.28 |
| WS | 44.45 | 37.17 | 37.88 | 43.06 | 44.61±0.04 |

Table 3: Average node-node similarity numerical evaluation on penultimate stage, corresponding graph sampling methods and base model

| Graph | GTI (penultimate stage) | Random Walk | Random Jump | Forest Fire | BaseModel |
|---|---|---|---|---|---|
| BA | 99.97% | 91.40% | 91.40% | 91.43% | 99.96±0.01% |
| ER | 99.64% | 99.79% | 99.81% | 99.81% | 99.66±0.03% |
| kronecker | 99.22% | 93.13% | 93.13% | 93.13 | 99.22±0.07% |
| WS | 99.99% | 99.90% | 99.90% | 99.91% | 99.99±0.00% |

## 4.5 NUMERICAL COMPARISON

Here, we use the Frobenius norm between adjacency matrices and average node-node similarity (Blondel et al., 2004) $sim(G'_{L-1}, G) = \frac{\sum_{i \in N} \sum_{j \in N} sim\left(n_i^{G'_{L-1}}, n_j^G\right)}{|N^G|}$ (where $\left|N^G\right|$ is the total number of nodes in original graph) to evaluate the similarity between generated stages, graph sampling methods and the original graph. Let $A = G - G'_{L-1}$, where $G'_{L-1}$ is the adjacency matrix of the penultimate stage, then $sim\left(n_i^{G'_{L-1}}, n_j^G\right)$ represents the node-node similarity with mismatch penalty (Heymans & Singh, 2003) between node $i \in G'_{L-1}$ and node $j \in G$. Here, the average node-node similarity indicates how $G'_{L-1}$ resembles $G$.

For comparison, we use the same model parameters as before to generate 100 examples of BA, ER, Kronecker, and WS networks. These newly generated networks serve as base models to help us evaluate the similarity between the penultimate stage of GTI and the original graph, as well as the effectiveness of the sampling methods. Unlike the sampling methods, the results of Table 2 and Table 3 demonstrate a consistent ability to retain topological features across a variety of different graph types.

## 5 DISCUSSION AND FUTURE WORK

This paper leveraged the success of GANs in (unsupervised) image generation to tackle a fundamental challenge in graph topology analysis: a model-agnostic approach for learning graph topological features. By using a GAN for each hierarchical layer of the graph, our method allowed us to reconstruct diverse input graphs very well, as well as preserving both local and global topological features when generating similar (but smaller) graphs. In addition, our method identifies important features through the definition of the reconstruction stages. A clear direction of future research is in extending the model-agnostic approach to allow the input graph to be directed and weighted, and with edge attributes.

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
