# OpenReview forum: "Graph Topological Features via GAN"
_ICLR.cc/2018/Conference — Reject_

### Official Review · AnonReviewer3 · 2017-11-27
**A badly written paper with questionable architechture design**

**Rating:** 3
**Confidence:** 4

**Review:**

The authors try to combine the power of GANs with hierarchical community structure detections. While the idea is sound, many design choices of the system is questionable. The problem is particularly aggravated by the poor presentation of the paper, creating countless confusions for readers. I do not recommend the acceptance of this draft.

Compared with GAN, traditional graph analytics is model-specific and non-adaptive to training data. This is also the case for hierarchical community structures. By building the whole architecture on the Louvain method, the proposed method is by no means truly model-agnostic. In fact, if the layers are fine enough, a significant portion of the network structure will be captured by the sum-up module instead of the GAN modules, rendering the overall behavior dominated by the community detection algorithm.

The evaluation remains superficial with minimal quantitative comparisons. Treating degree distribution and clustering coefficient (appeared as cluster coefficient in draft) as global features is problematic. They are merely global average of local topological features which is incapable of capturing true long-distance structures in graphs.

The writing of the draft leaves much to be desired. The description of the architecture is confusing with design choices never clearly explained. Multiple concepts needs better introduction, including the very name of their model GTI and the idea of stage identification. Not to mention numerous grammatical errors, I suggest the authors seek professional English writing services.

---

### Official Review · AnonReviewer1 · 2017-11-27
**Many missing pieces, likely overfitting, graph isomorphism not addressed**

**Rating:** 4
**Confidence:** 4

**Review:**

Quality: The work has too many gaps for the reader to fill in. The generator (reconstructed matrix) is supposed to generate a 0-1 matrix (adjacency matrix) and allow backpropagation of the gradients to the generator. I am not sure how this is achieved in this work. The matrix is not isomorphic invariant and the different clusters don’t share a common model. Even implicit models should be trained with some way to leverage graph isomorphisms and pattern similarities between clusters. How can such a limited technique be generalizing? There is no metric in the results showing how the model generalizes, it may be just overfitting the data.

Clarity: The paper organization needs work; there are also some missing pieces to put the NN training together. It is only in Section 2.3 that the nature of G_i^\prime becomes clear, although it is used in Section 2.2. Equation (3) is rather vague for a mathematical equation. From what I understood from the text, equation (3) creates a binary matrix from the softmax output using an indicator function. If the output is binary, how can the gradients backpropagate? Is it backpropagating with a trick like the Gumbel-Softmax trick of Jang, Gu, and Poole 2017 or Bengio’s path derivative estimator? This is a key point not discussed in the manuscript.
And if I misunderstood the sentence “turn re_G into a binary matrix” and the values are continuous, wouldn’t the discriminator have an easy time distinguishing the generated data from the real data. And wouldn’t the generator start working towards vanishing gradients in its quest to saturate the re_G output?

Originality: The work proposes an interesting approach: first cluster the network, then learning distinct GANs over each cluster. There are many such ideas now on ArXiv but it would be unfair to contrast this approach with unpublished work. There is no contribution in the GAN / neural network aspect. It is also unclear whether the model generalizes. I don’t think this is a good fit for ICLR.

Significance: Generating graphs is an important task in in relational learning tasks, drug discovery, and in learning to generate new relationships from knowledge bases. The work itself, however, falls short of the goal. At best the generator seems to be working but I fear it is overfitting. The contribution for ICLR is rather minimal, unfortunately.

Minor:

GTI was not introduced before it is first mentioned in the into.

Y. Bengio, N. Leonard, and A. Courville. Estimating or propagating gradients through stochastic neurons for conditional computation. arXiv:1308.3432, 2013.

---

### Official Review · AnonReviewer2 · 2017-11-28
**This paper presents a highly engineered approach for learning topological features of an input graph with GANs.  It is not clear why the approach works and under which conditions it could fail.**

**Rating:** 4
**Confidence:** 5

**Review:**

The proposed approach, GTI, has many free parameters: number of layers L, number of communities in each layer, number of non-overlapping subgraphs M, number of nodes in each subgraph k, etc.  No analysis is reported on how these affect the performance of GTI.

GTI uses the Louvain hierarchical community detection method to identify the hierarchy in the graph and METIS to partition the communities.  How important are these two methods to the success of GTI?

Why is it reasonable to restore a k-by-k adjacency matrix from the standard uniform distribution (as stated in Section 2.1)?

Why is the stride for the convolutional/deconvoluational layers set to 2 (as stated in Section 2.1)?

Equation 1 has a symbol E in it.  E is defined (in Section 2.2) to be "all the inter-subgraph (community) edges identified by the Louvain method for each hierarchy."  However, E can be intra-community because communities are partitioned by METIS.  More discussion is needed about the role of edges in E.

Equation 3 sparsifies (i.e. prunes the edges) of a graph -- namely $re_{G}$.  However, it is not clear how one selects a $re^{i}{G}$ from among the various i values.  The symbol i is an index into $CV_{i}$, the cut-value of the i-th largest unique weight-value.

Was the edge-importance reported in Section 2.3 checked against various measures of edge importance such as edge betweenness?

Table 1 needs more discussion in terms of retained edge percentage for ordered stages.  Should one expect a certain trend in these sequences?

Almost all of the experiments are qualitative and can be easily made quantitive by comparing PageRank or degree of nodes.

The discussion on graph sampling does not include how much of the graph was sampled.  Thus, the comparisons in Tables 2 and 3 are not fair.

The most realistic graph generator is the BTER model.  See http://www.sandia.gov/~tgkolda/bter_supplement/ and http://www.sandia.gov/~tgkolda/feastpack/doc_bter_match.html.

A minor point: The acronym GTI is never defined.

---

### Decision · Program_Chairs · 2018-01-29
**ICLR 2018 Conference Acceptance Decision**

**Decision:**

Reject

**Comment:**

The reviewers present strong concerns regarding presentation of the paper. The approach appears overly complex, some design choices are not clear and the experiments are not conducted properly. I recommend the authors to carefully go through the reviews.